# DeepEnhancerPPO: An Interpretable Deep Learning Approach for Enhancer Classification

**DOI:** 10.3390/ijms252312942

**Published:** 2024-12-02

**Authors:** Xuechen Mu, Zhenyu Huang, Qiufen Chen, Bocheng Shi, Long Xu, Ying Xu, Kai Zhang

**Affiliations:** 1School of Mathematics, Jilin University, Changchun 130012, China; m250921296@gmail.com (X.M.); shibc22@mails.jlu.edu.cn (B.S.); 2School of Medicine, Southern University of Science and Technology, Shenzhen 518055, China; zhenyuh19@mails.jlu.edu.cn (Z.H.); xul6@sustech.edu.cn (L.X.); 3College of Computer Science and Technology, Jilin University, Changchun 130012, China; 4School of Science, Southern University of Science and Technology, Shenzhen 518055, China; chenqf829@foxmail.com

**Keywords:** ResNet, transformer, PPO, interpretability, enhancer classification

## Abstract

Enhancers are short genomic segments located in non-coding regions of the genome that play a critical role in regulating the expression of target genes. Despite their importance in transcriptional regulation, effective methods for classifying enhancer categories and regulatory strengths remain limited. To address this challenge, we propose a novel end-to-end deep learning architecture named DeepEnhancerPPO. The model integrates ResNet and Transformer modules to extract local, hierarchical, and long-range contextual features. Following feature fusion, we employ Proximal Policy Optimization (PPO), a reinforcement learning technique, to reduce the dimensionality of the fused features, retaining the most relevant features for downstream classification tasks. We evaluate the performance of DeepEnhancerPPO from multiple perspectives, including ablation analysis, independent tests, assessment of PPO’s contribution to performance enhancement, and interpretability of the classification results. Each module positively contributes to the overall performance, with ResNet and PPO being the most significant contributors. Overall, DeepEnhancerPPO demonstrates superior performance on independent datasets compared to other models, outperforming the second-best model by 6.7% in accuracy for enhancer category classification. The model consistently ranks among the top five classifiers out of 25 for enhancer strength classification without requiring re-optimization of the hyperparameters and ranks as the second-best when the hyperparameters are refined. This indicates that the DeepEnhancerPPO framework is highly robust for enhancer classification. Additionally, the incorporation of PPO enhances the interpretability of the classification results.

## 1. Introduction

The spatiotemporal control of gene expressions ensures cell differentiation and plasticity, which is essential for the development of a single-cell embryo into a complex organism [1]. Enhancers, a class of genomic cis-regulatory elements, act as key regulators of gene expression by binding with transcription factors (TFs) and complexes that control gene expressions [2]. These elements can be located both upstream and downstream of their target genes, including within introns, significantly influencing transcriptional burst frequency [3,4]. Despite their crucial role in transcription regulation, the discovery and classification of enhancers remain challenging.

Traditionally, enhancers have been identified through experimental approaches that predict enhancers by capturing the molecular features associated with them, such as transcription factor (TF) binding sites, histone modifications, and nucleosome-depleted regions (NDRs) [5]. For instance, Heintzman and Ren identified novel enhancers by examining binding affinities to transcription factors like P300 [6]. Yan et al. discovered an intronic enhancer regulating RANKL expression in osteocytic cells through chromatin landscape analysis [7]. DNase-seq (DNase I hypersensitive sites sequencing) facilitates the identification of enhancers by capturing NDRs [8]. Shyamsunder et al. used DNase-seq and CRISPR/Cas9 technology to uncover a novel enhancer of CCAAT/enhancer binding protein ϵ (CEBPE) [9]. However, these experimental methods are time-consuming and labor-intensive and may fail to detect enhancers due to their condition-specific activities [10].

To address these challenges, various computational methods have been developed for enhancer identification. Machine learning approaches have shown promise in tackling this challenge. Firpi et al. introduced CSI-ANN, an artificial neural network-based algorithm that efficiently extracts enhancer sequence features and accurately detects novel enhancers [11]. EnhancerFinder combines multiple learning kernels based on evolutionary conservation patterns, sequence motifs, and cell type-specific functional information to detect and characterize enhancers [12]. Additionally, a random forest (RF) model trained on chromatin status data has been used to construct enhancer maps across multiple cell types [13].

Recent advancements in deep learning have further advanced the state-of-the-art of enhancer detection. EnhancedDBN employs a deep belief network (DBN) for this task [14]. Liu et al. used the pseudo-k-set nucleotide composition (PseKNC) algorithm [15] to derive sequence features and proposed a two-layer classifier, iEnhancer-2L, for both enhancer detection and strength determination. Building on this, Jia et al. [16] developed EnhancerPred by applying a two-step wrapper feature selection algorithm to refine features from bi-profile Bayes and PseKNC, significantly improving the Matthews Correlation Coefficient (MCC) for the two layers of iEnhancer-2L.

Further advancements in deep learning have utilized convolutional operations for enhancer detection. Xu et al. introduced Enhancer-CNN, a convolutional neural network model that treats DNA sequences as 1-D images, achieving high accuracy in enhancer prediction across different cell types [17]. Nguyen et al. integrated one-hot encoding and statistical k-mer descriptors with convolutional neural networks (CNNs) to create iEnhancer-ECNN, demonstrating enhanced prediction performance on independent test datasets [18]. Hybrid models combining CNNs and bidirectional long short-term memory (BiLSTM) networks have also shown promise by capturing both local sequence features and long-range dependencies, significantly improving prediction performance [19,20,21]. Additionally, Le et al. proposed BERT-Enhancer, a model that leverages the transformer-based BERT architecture to capture complex interactions within enhancer sequences, demonstrating superior performance in identifying enhancers across various genomic contexts [22].

Despite these advancements, there is still a need for improving the performance in terms of both accuracy and result interpretability in enhancer detection and classification. Existing methods generally fall short in achieving high accuracy and are computationally expensive due to model complexity, often ignoring model interpretability. To overcome these limitations, we propose a novel computational method that combines a simplified ResNet18-style [23] and transformer [24] feature extraction with reinforcement learning. Our approach leverages the advantages of both deeper convolutional operations and transformer-based temporal interactions to extract complex DNA features. Moreover, by incorporating reinforcement learning, specifically proximal policy optimization (PPO) [25], our model effectively reduces feature dimensionality and enhances interpretability.

Our approach treats each DNA sequence as a natural language sentence, employing k-mer techniques to segment the sequence into k-mer words. Each k-mer word is then embedded into a dense feature space. We utilize a residual network (ResNet) architecture, consisting of one-dimensional convolutional residual blocks, to extract local and hierarchical features from the embedded k-mer representations. In parallel, a transformer module captures long-range dependencies and temporal interactions through multi-head self-attention mechanisms. The outputs from these two feature extraction pathways are concatenated to create a fused feature representation. The Proximal Policy Optimization (PPO) reinforcement learning component is subsequently used to evaluate the fused features, making automated decisions about which features to retain or discard, thereby enhancing the informativeness and relevance of the features for downstream enhancer classification tasks. Our model, termed DeepEnhancerPPO, demonstrates significant improvements across multiple performance metrics, including accuracy, Matthews Correlation Coefficient (MCC), and Area Under the Curve (AUC), when compared to 24 state-of-the-art methods. Furthermore, we conduct an extensive evaluation of the PPO-based feature reduction module, comparing it against seven classical feature selection methods and four state-of-the-art reinforcement learning algorithms. Interpretability analyses are also performed to identify the most important features and validate meaningful learning using established biological features associated with enhancer activity, thus highlighting the efficacy and advantages of PPO within our framework.

In summary, our integrated approach not only achieves superior performance in enhancer classification but also offers enhanced interpretability and computational efficiency.

## 2. Results


In this section, we systematically evaluate the parameters of our proposed model, DeepEnhancerPPO, and compare its performance with 24 existing models on the same independent datasets for two distinct enhancer classification tasks: category prediction and strength prediction. We divided 10% of the 2968 training DNA sequences into a validation set using a random seed of 75, with the remaining sequences used for training the models. We first explore various potential representation strategies and hyperparameters for DeepEnhancerPPO, including the choice of k in k-mer, batch size, and the dimensionality of the embedding space, using the validation dataset. Subsequently, we conduct ablation studies to assess the impact of different modules on the overall performance of DeepEnhancerPPO, helping us identify the components that contribute most significantly to the model’s efficacy.

During the training phase, we monitor the accuracy of DeepEnhancerPPO on the validation dataset to identify the optimal model parameters for the first task: enhancer category classification. These optimized parameters are then applied to make predictions on the independent test dataset. For the strength classification task, we evaluate two different hyperparameter combinations: one set identical to those used in the category classification task, which demonstrates the generalizability of DeepEnhancerPPO across different hyperparameter configurations, and another set refined from various combinations of hyperparameters (k-mer, embedding size, and batch size) to highlight the superior performance of DeepEnhancerPPO when compared to other state-of-the-art (SOTA) models.

To further demonstrate the capabilities of DeepEnhancerPPO, we conduct additional experiments that explore its performance from multiple perspectives. These include evaluating its performance on another benchmark dataset using the same hyperparameters to demonstrate the generalizability of DeepEnhancerPPO, assessing the superior performance of PPO within DeepEnhancerPPO compared to seven classical feature selection algorithms and four SOTA deep reinforcement learning (DRL) algorithms, and providing a comprehensive interpretability analysis of DeepEnhancerPPO due to the inclusion of the feature reduction module, PPO.

All experiments and analyses were conducted using a random seed of 42, a commonly used value for ensuring reproducibility in computational experiments [26].

### 2.1. Determination of K-mers Using the Available Learning Embedding

K-mers are introduced to split a DNA sequence into smaller subsequences, known as k-mer words. These k-mer words are then converted into numerical representations according to *DNA2Num*. The choice of k is critical, as different downstream tasks may require different k-mer values to achieve optimal performance. Previous studies have demonstrated that there is no universally best k-mer value for all tasks [27].

To determine the optimal k-mer value for our enhancer category classifier, we conducted experiments on the validation dataset. Figure 1 illustrates the accuracy of the model for different k-mer values, ranging from 1-mer to 6-mer. Our results show that the best performance, with an accuracy of 85.19%, is achieved using 4-mers, outperforming the second-best value, 5-mer, by 1.69% (accuracy of 83.50%). Therefore, we select 4-mer as the optimal value for splitting DNA sequences and encoding the k-mer word embeddings in the subsequent experiments.

### 2.2. A Comprehensive Exploration of K-mer Word Embeddings Based on Two Pre-Trained Models

As mentioned earlier, we have determined that 4-mers provide the best performance compared to other k-mer values, and thus, we use 4-mers to split DNA sequences. These sequences are then input into an embedding space to learn their representations through the available learning method. In this section, we further investigate the integration of pre-trained embeddings, specifically DNA2Vec [28] and DNABERT [29], with the learned embeddings. Our goal is to assess whether incorporating superior pre-trained embedding techniques can improve the performance of DeepEnhancerPPO.

We explore two strategies for combining the learned embeddings with pre-trained embeddings: independent feature extraction and shared feature extraction. These strategies are evaluated to determine whether the inclusion of pre-trained embeddings enhances the performance of DeepEnhancerPPO.

#### 2.2.1. Independent Feature Extraction of DNA2Vec Embedding with Existing Learned Embedding

We first incorporate DNA2Vec embeddings, with k values ranging from 3 to 8, into DeepEnhancerPPO independently from the original learned embedding and feature extraction path. A separate feature extraction process, involving ResNet for local and hierarchical feature extraction and a Transformer for capturing long-range contextual features, is applied to the DNA2Vec embeddings. The deep features from the four feature extraction paths are then concatenated to form a final fused feature, which is subsequently reduced in dimensionality by PPO for enhancer category classification (see Figure 2A, labeled ’separated-DNA2Vec’) .

The results indicate that the addition of pre-trained embeddings does not enhance the performance of DeepEnhancerPPO on the validation dataset when added independently in this manner. The highest accuracy achieved by this strategy is 78.11%, corresponding to the combination of 4-mer learned embedding with 6-mer DNA2Vec embedding—this represents a 7.08% decrease compared to the 85.19% accuracy obtained using only the 4-mer learned embedding (Figure 1). We speculate that this decrease may be attributed to the overly complex feature extraction and fusion mechanisms. Consequently, we further optimize the embedding combination strategy to compare against the performance of the solely learnable 4-mer embedding.

#### 2.2.2. Shared Feature Extraction for DNA2Vec and DNABERT Embeddings with Existing Learned Embedding

In the previous section, we used an independent extraction strategy for pre-trained embedding paths. Here, we investigate a fusion approach prior to feature extraction, without adding additional feature extraction paths. We evaluate two potential combination strategies: direct concatenation and an attention-weighted mechanism. First, we determine the better fusion strategy by introducing DNA2Vec to DeepEnhancer, then we consider using a more sophisticated pre-trained embedding, comparing it with the solely learned 4-mer embedding to assess the potential benefits of introducing pre-trained embeddings. To ensure compatibility, a truncation strategy is employed to maintain consistent sequence lengths for different k-mers from DNA2Vec (k ranging from 3 to 8) when concatenating with the 4-mer learned embeddings.

For the direct concatenation strategy, the two different k-mer word embeddings are concatenated along the embedding feature dimension. In the attention-weighting mechanism, we use an attention module where the learned embeddings serve as weights and the DNA2Vec embeddings as values. The attention output is then concatenated with the learned 4-mer embeddings to form the final fused embedding. Regardless of the fusion strategy, a linear projection is applied to map the combined embedding dimension to 128. The results for both fusion strategies are shown in Figure 2B and C, respectively. Our findings indicate that the direct concatenation strategy outperforms the attention-weighting mechanism, as the majority of combinations involving various k-mers of DNA2Vec with the learned 4-mer embedding exceed 80.00% in accuracy on the validation dataset, as shown in Figure 2C.

However, we also note that although the direct concatenation strategy for the pre-trained DNA2Vec embeddings with the solely learned 4-mer embedding achieves higher accuracy compared to the attention-weighting mechanism, both methods reach a maximum accuracy of 83.50%, which is still below the 85.19% accuracy achieved using only the learned 4-mer embedding (Figure 1). We speculate that this outcome may be due to the limitations of DNA2Vec as a pre-trained embedding technology. Therefore, we select the better-performing fusion strategy—direct concatenation—and combine it with a more advanced pre-trained embedding method, DNABERT [29], using k-mer values ranging from 3 to 6. We aim to evaluate whether a SOTA pre-trained embedding can enhance the performance of DeepEnhancer, as depicted in Figure 2D.

Regrettably, even when leveraging the SOTA DNABERT pre-trained embedding alongside the solely learned 4-mer embedding, we observe no improvement in performance. The accuracy of both the fused pre-trained DNABERT 4-mer embedding and the solely learned 4-mer embedding is 85.19% on the validation dataset. Moreover, the introduction of the DNABERT pre-trained embedding significantly increased the training time for DeepEnhancerPPO—from approximately 8 h using only the learned 4-mer embedding to around 42 h. This is primarily due to the larger embedding dimension of DNABERT (768), compared to the 128-dimensional solely learned 4-mer embedding, representing a six-fold increase in computational load, which is clearly unacceptable given the lack of performance improvement.

These results indicate that these fusion strategies, whether using DNA2Vec or DNABERT, do not positively impact the performance of DeepEnhancerPPO. We hypothesize that the depth of feature extraction layers in DeepEnhancerPPO, combined with the relatively moderate sample size for enhancer classification, hinders the effective fine-tuning of pre-trained embeddings [30]. Therefore, we decided to utilize only the learned 4-mer embedding strategy, which demonstrated the best performance on the validation dataset (Figure 1), and proceed with this representation strategy for all subsequent experiments.

### 2.3. Determination of Hyperparameters

In this section, we conduct a comprehensive experiment to identify the optimal combination of the hyperparameters for the embedding dimension and batch size. We define two pools for the embedding dimension and batch size, consisting of commonly used values: embeddingdimensionset=[64,128,256,300] and batchsizeset=[32,64,128]. We explore all possible combinations of these hyperparameters, and the results are presented in Figure 3.

The results indicate that two parameter combinations exhibit particularly strong performance on the validation set: an embedding dimension of 128 with a batch size of 64, and an embedding dimension of 256 with a batch size of 64. To determine the most suitable combination, we conducted a hypothesis test using McNemar’s test [31], which is appropriate for evaluating the significant differences in categorical outcomes between two related groups, on the two sets of predictions from the validation dataset. The *p*-value obtained from McNemar’s test was 0.396, indicating no significant difference between the two combinations.

Furthermore, we recorded the computational time for both combinations. For the embedding dimension of 128 with a batch size of 64, the time cost was 34,946.38 s, whereas for the embedding dimension of 256 with a batch size of 64, it was 35,834.85 s. Considering the limitations of our experimental setup, which consists of a single NVIDIA A800 80 G GPU, and more importantly, given the lack of significant performance differences between the two combinations, we opted to use the combination of an embedding dimension of 128 and a batch size of 64 for subsequent experiments. This choice strikes a balance between computational efficiency and model performance, ensuring robust results on the validation dataset.

### 2.4. Ablation Study

We explore the contributions of each module within DeepEnhancerPPO, focusing on three main components: ResNet, Transformer (both responsible for feature extraction from different perspectives), and PPO (used for feature dimension reduction). The complete results are presented in Figure 4.

By comparing Figure 1, which shows the performance of DeepEnhancerPPO on the validation dataset, with Figure 4, we gain significant insights into the impact of each module. The ResNet module demonstrates the most substantial benefit for DeepEnhancerPPO, evidenced by a 34.68% decrease in validation accuracy upon its removal (from 85.19% to 50.51%). This aligns with previous studies indicating that convolutional operations are highly suitable for processing biological sequences [32].

The PPO feature reduction module is the second most impactful component, with a 12.46% drop in validation accuracy when excluded (from 85.19% to 72.73%). This indicates the importance of feature reduction in enhancing model performance by retaining only the most relevant features.

Although the Transformer module has the least impact compared to ResNet and PPO, its removal still results in a 9.1% decrease in the validation accuracy (from 85.19% to 76.09%). This demonstrates that capturing long-range dependencies and temporal interactions remains crucial for the overall performance of DeepEnhancerPPO.

These observations highlight that while all three modules—ResNet, Transformer, and PPO—are essential for the success of DeepEnhancerPPO, their relative importance varies. ResNet represents the most significant contributor, followed by PPO, with the Transformer module also contributing positively to enhancer category classification.

### 2.5. Comparison of DeepEnhancerPPO with 24 State-of-the-Art Models

In this section, we present a comparative analysis of DeepEnhancerPPO against 24 existing state-of-the-art models. The parameters of DeepEnhancerPPO are optimized as described in previous sections. Specifically, we utilize 4-mers to segment each DNA sequence, and an embedding module is applied to represent each 4-mer word. Feature extraction is subsequently performed using ResNet and Transformer modules to capture local, hierarchical, and long-range context features. The hyperparameters of DeepEnhancerPPO, including a batch size of 64 and an embedding dimension of 128, are selected for these experiments.

Notably, during the comparison phase, we first employ the same combination of hyperparameters for the second classification task (enhancer strength classification) as used in the first task (enhancer category classification) to demonstrate the generality of DeepEnhancerPPO across different tasks. Concurrently, we also refine the hyperparameter combination for the strength classification task following the same selection process used earlier to compare DeepEnhancerPPO with the 24 state-of-the-art models, thereby demonstrating its potential effectiveness for multiple tasks. The random seed for all experiments is set to 42, which is a commonly chosen seed value to ensure reproducibility [26]. During training, the optimal model parameters are recorded based on performance on the validation dataset.

#### Performance of DeepEnhancerPPO Compared to 24 State-of-the-Art Models on the Independent Test Dataset for Two Enhancer Classification

The performance metrics Sn (Sensitivity) and Sp (Specificity) measure different aspects of a binary classification model, where an improvement in one metric may come at the expense of the other [33]. Overall prediction performance metrics such as Acc (Accuracy), MCC (Matthews Correlation Coefficient), and AUC (Area Under the Curve) are used for a fair comparison of the two enhancer classification tasks: category and strength. All algorithms, including DeepEnhancerPPO, are evaluated based on their prediction performances on independent datasets that were not involved in the model training process, to assess their generalization capabilities on future samples, as shown in Table 1 and Table 2.

Table 1 provides a comparative analysis of DeepEnhancerPPO and the 24 existing state-of-the-art models on the independent test dataset for the enhancer category classification task. The results clearly indicate that DeepEnhancerPPO outperforms all 24 existing models in terms of ACC, MCC, and AUC. Notably, the accuracy of DeepEnhancerPPO is 6.7% higher than the second-best model, ADH-Enhancer [51]. The MCC and AUC metrics both have significant improvement, further establishing the superiority of DeepEnhancerPPO.

As previously stated, we first employ the same combination of hyperparameters used in the enhancer category classification task to train DeepEnhancerPPO for the second task, namely enhancer strength classification, in order to demonstrate its stable performance across different tasks. This version is referred to as ‘DeepEnhancerPPO’. Subsequently, we refine the hyperparameters—specifically, the value of k for k-mer, the embedding dimension, and the batch size—for the second classification task to unlock the potential capabilities of DeepEnhancerPPO, which we refer to as ’DeepEnhancerPPO-Refined’. The results for both versions are presented in Table 2.

We first examine the performance of the non-refined version, ‘DeepEnhancerPPO’. Although it does not outperform all 25 algorithms across the three evaluation metrics (ACC, MCC, and AUC), it consistently ranks among the top five classifiers. This demonstrates that DeepEnhancerPPO exhibits robustness and versatility without significant shortcomings for both enhancer classification tasks.

Next, we turn our attention to ‘DeepEnhancerPPO-Refined’, which is trained using refined hyperparameters specifically for the k-mer value, embedding dimension, and batch size (detailed hyperparameter settings can be found in Appendix C). We observe an improvement of 6.5% in accuracy, positioning it as the second-best classifier for the enhancer strength classification task. This improvement indicates that DeepEnhancerPPO has significant potential for DNA sequence classification.

In conclusion, when combined with the interpretability analysis, DeepEnhancerPPO stands out as a powerful model compared to the existing 24 enhancer detection methods, with a strong potential for diverse real-world applications.

### 2.6. Further Performance Analysis of DeepEnhancerPPO on Another Benchmark for Enhancer Category Classification

In addition to the comprehensive comparison presented earlier, we further evaluate DeepEnhancerPPO on another recent benchmark for enhancer category classification, namely ‘The Human Enhancers Ensembl Dataset’ [52]. This dataset was initially constructed from human enhancers in the FANTOM5 project [53], accessed through the Ensembl database [54]. The negative sequences were randomly generated by Grešová et al. [52] from the human genome GRCh38, with lengths matching those of positive sequences and no overlap between them. The dataset originally included a training set and a test set, containing DNA sequences of varying lengths, ranging from less than 10 bp to more than 550 bp. We split the original training dataset into a sub-training dataset (used for training DeepEnhancer) and a validation dataset (used to store the optimal model parameters), with a 9:1 ratio. This resulted in 35,522 sequences in the sub-training dataset and 3947 sequences in the validation dataset. The median sequence length of the sub-training dataset is 239, while the median length of the validation dataset is 233. Therefore, during training, we set the maximum sequence length to 239, based on the median length of the sub-training dataset. The original test dataset contains 10,137 sequences, with a median length of 236.

For this experiment, we use the same hyperparameters as those used in the first task (enhancer category classification) to train DeepEnhancer and evaluate its performance on the test datasets, as shown in Table 3.

From the results presented in Table 3, it is evident that DeepEnhancerPPO achieves the best performance, with an accuracy superior to the second-best model by 3.72%. Notably, this result is obtained without further refining the hyperparameters of DeepEnhancerPPO. This finding reinforces the potential of DeepEnhancerPPO in DNA sequence classification, particularly for enhancer category classification.

### 2.7. A Deeper Analysis of the Feature Reduction Module of DeepEnhancerPPO

In this section, we explore the advantages of the feature reduction module in DeepEnhancerPPO, specifically the Proximal Policy Optimization (PPO) component from two aspects as follows: (1) comparing the performance of PPO with seven classical feature selection algorithms and four SOTA DRL models on the independent dataset, and (2) evaluating the interpretability of DeepEnhancerPPO enabled by the PPO-based feature reduction.

#### 2.7.1. The Best Feature Reduction Choice for DeepEnhancerPPO: PPO

We first conduct a comprehensive comparison of the feature reduction strategies in DeepEnhancerPPO, using PPO, with 11 alternative feature reduction approaches. These include seven classical feature selection algorithms: CIFE [55], CMIM [56], DISR [57], ICAP [58], JMI [59], MIM [60], and MRMR [61], as well as four SOTA DRL models: A2C [62], DDPG [63], DPO [64], and SAC [65]. We evaluate these algorithms on both enhancer category and enhancer strength classification tasks to demonstrate that PPO is the best choice within our framework.

To ensure a fair comparison, we use different sub-feature dimensions for the classical feature selection algorithms. Given that the total concatenated feature dimension of DeepEnhancerPPO (from ResNet and Transformer) is 192, we test three common sub-feature numbers: 32, 64, and 128. Figure 5 and Figure 6 show the comparison of these strategies for enhancer category and enhancer strength classification, respectively.

The results, as shown in Figure 5 and Figure 6, demonstrate that DeepEnhancerPPO consistently outperforms all 11 other feature selection algorithms, irrespective of the classification task (category or strength). Specifically, the feature reduction strategy using PPO (indicated by the gray bar) achieves the highest performance across all DRL-based feature reduction strategies and classical sub-feature dimensions (32, 64, and 128), as shown in Figure 5A–C and Figure 6A–C. This confirms that PPO is the best choice for feature reduction in DeepEnhancerPPO, as it consistently leads to superior results when combined with the concatenated features from ResNet and Transformer.

Furthermore, we observe that the DRL-based feature reduction strategies, including A2C, DDPG, DPO, and SAC, provide a more stable performance than the classical feature selection methods. For instance, some of the classical algorithms, such as MIM-32 (Figure 5A) and DISR-64 (Figure 5B), exhibit negative MCC values, indicating poor performance. In contrast, the DRL-based strategies maintain a moderate but stable performance across different configurations. This suggests that DRL is a promising approach for feature reduction in deep learning frameworks, due to its end-to-end nature, which allows for more flexible and adaptive feature selection compared to the knowledge-based strategies of classical feature selection.

#### 2.7.2. Interpretability Analysis of DeepEnhancerPPO Based on PPO

The previous sections have demonstrated the superior performance of DeepEnhancerPPO in both enhancer category and strength classification tasks, with PPO emerging as the most effective feature reduction strategy for the concatenated feature set. In this section, we conduct a comprehensive interpretability analysis to further investigate the role of PPO within DeepEnhancerPPO. This analysis is approached from two perspectives as follows: first, we leverage PPO’s inherent capabilities to identify which features in the fused feature set are most crucial for enhancer classification in both tasks (enhancer category and enhancer strength classification). Second, we conduct a correlation analysis between the features before and after masking by PPO, comparing these with reference features based on biological knowledge of enhancer activity. This helps to determine whether PPO has learned meaningful features related to enhancer activity, specifically whether the masked features direct the model towards relevant biological insights.

We begin by discussing the first aspect in detail: the impact of PPO’s feature masking on enhancer classification performance. After training on the training dataset and selecting the optimal model parameters based on validation performance, we applied DeepEnhancerPPO to predict enhancers in an independent test dataset for both classification tasks. Simultaneously, we visualized the fusion feature masking determined by PPO during prediction, as shown in Figure 7 (for enhancer category classification) and Figure 8 (for enhancer strength classification). The x-axis of each heatmap represents the 192 dimensions of the masked fusion features, where the first 64 entries correspond to the ResNet output, and the remaining entries are derived from the Transformer module. The y-axis represents the total number of batches processed for each task.

From Figure 7 and Figure 8, we observe that the top 64 features consistently show higher intensity, suggesting that these features are of greater importance. This pattern indicates that ResNet plays a more pivotal role in the DeepEnhancerPPO framework. This finding is consistent with our earlier ablation study, where removing ResNet resulted in a substantial decrease in validation accuracy, from 85.19% to 50.51%, a reduction of 34.68%. More importantly, the PPO-based feature reduction module automatically identifies this key feature selection pattern, without requiring explicit biological knowledge to guide it. This further underscores the value of PPO in facilitating meaningful feature selection, illustrating the strength of the design.

Next, we further explore the interpretability of the post-masked fusion features by PPO, correlating them with reference features derived from biological knowledge. Well-established research indicates that ‘GC Content’, which refers to the proportion of guanine (G) and cytosine (C) nucleotides, and ‘CpG Islands’, which refer to the number of CpG dinucleotides, are strongly associated with enhancer activity. Both of these features are known to be significantly enriched in enhancer sequences [66]. Therefore, we combine these two features with ‘nucleotide composition’, which refers to the frequency of adenine (A), thymine (T), guanine (G), and cytosine (C) nucleotides, to form our reference features. We then apply principal component analysis (PCA) [67] to reduce the dimensionality of both the pre-masked and post-masked fusion features generated by PPO, ensuring that they are aligned with the reference features in terms of dimensionality (i.e., reduced to six dimensions). This allows us to calculate the Pearson correlation [68], as shown in Figure 9.

In Figure 9, the y-axis represents the Pearson correlation coefficient between the reduced features of the pre-masked and post-masked fusion features and the reference features. The x-axis corresponds to the sample index from the independent test dataset, where the top 200 samples are labeled as enhancers (i.e., labeled 1) and the bottom 200 samples as non-enhancers (i.e., labeled 0). Interestingly, the correlation between the pre-masked fusion features and the reference features shows no clear pattern across the enhancer and non-enhancer samples. In contrast, the post-masked fusion features exhibit a distinct and opposite pattern. Specifically, the correlation for the top 200 samples remains above the zero axis, indicating a positive correlation with the reference features, while the correlation for the bottom 200 samples falls below the zero axis, indicating a negative correlation with the reference features. This observation is significant because the reference features are closely associated with enhancer activity, and it is expected that the non-enhancer sequences (the bottom 200 samples) would exhibit a negative correlation with these reference features.

Overall, both the heatmap analysis, which identifies significant features, and the correlation analysis, which demonstrates the consistency of feature learning and masking, provide strong evidence of the effectiveness of PPO within DeepEnhancerPPO. These results underscore the interpretability of the model and guide future refinement and application of the classifier.

## 3. Materials and Methods

### 3.1. Benchmark Datasets

In this study, we evaluate the performance of DeepEnhancerPPO using two benchmark datasets. The primary focus is on the first benchmark due to the extensive number of classifiers used, while the second benchmark serves as an additional validation to further demonstrate the generalizability and robustness of DeepEnhancerPPO. The first benchmark, originally constructed by Liu et al. [15], includes both training and independent test datasets. As mentioned earlier, our evaluation involves 24 state-of-the-art models, all assessed on the same datasets, ensuring a fair comparison. Each model is trained on the same training dataset and evaluated on the same independent test set. Specifically, the training dataset, denoted as *S*, consists of 2968 DNA sequences: 742 strong enhancer sequences, 742 weak enhancer sequences, and 1484 non-enhancer sequences, as annotated in the original study. The independent test dataset, which does not overlap with the training dataset, contains 100 strong enhancer sequences, 100 weak enhancer sequences, and 200 non-enhancer sequences. All sequences in both datasets are 200 base pairs (bp) in length. In the first task, both strong and weak enhancers are treated as positive samples, while non-enhancer sequences are considered negative samples. In the second task, strong enhancers are classified as positive samples and weak enhancers are treated as negative samples.

The second benchmark for enhancer sequence classification involves both enhancer and non-enhancer sequences, specifically for the enhancer category classification task. The sequences in this dataset vary in length, and detailed information regarding this dataset is provided in Section 2.6.

### 3.2. Evaluation Metrics

To assess the performance of DeepEnhancerPPO as well as the other 24 programs, we adopt five commonly-used classification performance metrics, identical to those used in previous studies. These metrics include accuracy (ACC), sensitivity (SN), specificity (SP), Matthews correlation coefficient (MCC), and the area under the ROC curve (AUC).

ACC measures the percentage of correctly classified instances out of all instances. SN, also known as recall, represents the true positive rate, representing the proportion of positive instances correctly predicted. SP, or the true negative rate, denotes the ratio of true negatives to all negative outcomes. The MCC measures the correlation between the actual and predicted values of the instances. The AUC represents the model’s ability to discriminate between positive and negative instances. Generally, higher values for these metrics indicate superior model performance [69]. The mathematical expressions for these metrics are as follows:(1)$$\mathrm{ACC}=\frac{TP+TN}{TP+TN+FP+FN}$$
(2)$$\mathrm{SN}=\frac{TP}{TP+FN}$$
(3)$$\mathrm{SP}=\frac{TN}{TN+FP}$$
(4)$$\mathrm{MCC}=\frac{TP\times TN-FP\times FN}{\sqrt{\left(TP+FP)(TP+FN)(TN+FP)(TN+FN\right)}}$$
where *TP, TN, FP,* and *FN* represent the number of true positives, true negatives, false positives, and false negatives, respectively.

### 3.3. Feature Representation of DNA Sequences

The effective representation of DNA sequences is crucial for accurately distinguishing between enhancers and non-enhancers, as well as between strong and weak enhancers. In this study, we conceptualize DNA sequences as a form of natural language, transforming nucleotide sequences into numerical vectors that serve as inputs to DeepEnhancerPPO.

#### 3.3.1. K-mer Encoding

A nucleotide sequence is typically represented as a string of characters (A, C, G, and T). We employ a sliding window technique to transform each sequence into a series of overlapping k-mers with a step size of 1 (see Figure 10). For a sequence of length *N* and a k-mer length *k*, this approach generates N−k+1 k-mers. These k-mers are used to create a vocabulary, termed *DNA2Num*, built from the training dataset. Each k-mer is assigned a unique integer identifier, with the most frequent k-mers in the training dataset assigned the smallest identifiers, beginning with 2. Special identifiers ‘<unk>’ and ‘<pad>’ are assigned to 0 and 1, respectively. The ‘<unk>’ identifier addresses unknown sequences, while the ‘<pad>’ identifier is used to ensure consistent sample lengths.

#### 3.3.2. Vector Representation Through Embedding and Its Advantages

Transforming DNA sequences into integer sequences facilitates their embedding into a dense vector space, akin to word embedding in natural language processing [70]. This embedding layer, initially randomly initialized, is optimized during model training, capturing essential sequence representations relevant to enhancer classification.

This embedding approach effectively reduces the sparsity and dimensionality of the input feature space, making it more suitable for processing by advanced deep learning architectures such as ResNet and Transformer, which are integral components of DeepEnhancerPPO. Additionally, we compare pre-trained embeddings with the above learnable embeddings to determine the most effective representation, as discussed in Section 2.2.

### 3.4. Residual Module for Local and Hierarchical Feature Extraction

Residual Networks (ResNets) leverage their architectural innovations to enable the effective training of significantly deeper convolutional neural networks. By introducing skip connections, ResNets address the vanishing gradient problem [23], allowing gradients to propagate more efficiently during training. This structure supports the extraction of complex and hierarchical features from DNA sequences, thereby enhancing the detection and characterization of enhancers.

#### 3.4.1. Adaptation to DNA Sequences and Extracting Local Features

ResNet has been tailored for DNA sequence analysis by employing one-dimensional convolutional layers that slide through the sequence to detect local patterns. These layers capture motifs and short-range dependencies within the DNA sequences [71]. The core component of the ResNet architecture is the residual block, which incorporates skip connections to bypass one or more layers. These skip connections enable the network to maintain the integrity of the gradient flow, thereby supporting the construction of deeper network architectures.

In simpler terms, ResNet functions like a magnifying glass that not only observes individual nucleotides in a DNA sequence but also recognizes motifs and patterns by examining multiple layers of information simultaneously.

For detailed mathematical formulations and technical aspects of the ResNet architecture, please see Appendix A.

#### 3.4.2. Enhancing Depth for Feature Hierarchy

The integration of skip connections facilitates the construction of deeper network architectures, fostering the learning of abstract representations of input sequences. Consequently, deeper layers can identify subtle and complex relationships, ranging from simple motifs recognized in the initial layers to intricate interactions discerned in subsequent layers. This hierarchical representation is crucial for enhancer detection, where the significance of a sequence is contingent upon the arrangement of its elements at various scales [72].

Overall, incorporating ResNet into our framework for DNA sequence analysis provides a robust mechanism for deriving meaningful and complex features from biological sequences. This capability significantly augments the efficacy of subsequent classification tasks (see the ablation study), such as differentiating between enhancer and non-enhancer regions or distinguishing between strong and weak enhancers.

### 3.5. Transformer for DNA Sequence Feature Extraction

The transformer, originally developed for natural language processing tasks, utilizes a mechanism known as self-attention. This mechanism enables the model to weigh the importance of different components extracted from the input sequence [24]. For DNA sequence analysis, this capability is particularly advantageous as it allows the model to focus on crucial regulatory elements that might be widely spaced. Consequently, we employ the multi-head self-attention component of the transformer to extract both upstream and downstream temporal features from DNA sequences.

#### 3.5.1. Incorporation of Positional Information

To effectively utilize the multi-head self-attention, it is essential to incorporate positional information into the k-mer word embeddings. Following the approach outlined by Vaswani et al. [24], we encode positional information using sine and cosine functions. These functions provide a unique, continuous representation of each position in the sequence, distinguishing k-mers at different positions even if they are identical.

In simple terms, positional encoding acts like a unique address system, ensuring that the model recognizes the specific location of each k-mer in the DNA sequence.

For the detailed mathematical formulations and technical aspects of positional encoding, please refer to Appendix B.

#### 3.5.2. Multi-Head Self-Attention Mechanism

The multi-head self-attention mechanism enables the model to attend to information from different representation subspaces at various sequence positions simultaneously. This mechanism enhances the model’s ability to capture complex dependencies and interactions within the sequence. Each attention head processes information independently, allowing the model to integrate various aspects of the sequence context.

In simple terms, the multi-head self-attention acts like multiple spotlight beams, each focusing on different parts of the DNA sequence to gather comprehensive information about regulatory elements.

For the detailed mathematical formulations and technical aspects of the multi-head self-attention mechanism, please refer to Appendix B.

#### 3.5.3. Layer Composition and Normalization

After the multi-head attention mechanism processes the input, the concatenated output from these multiple heads is passed through a feed-forward network (FFN) with a non-linear activation function, typically ReLU, to enhance its expressiveness [24]. This is followed by a residual connection and layer normalization, which help capture different scales of feature information, from simple motifs to complex interactions, and stabilize the training process, respectively.

These operations form an encoder block, which can be stacked multiple times to capture subtle patterns and long-range interactions within the sequence [24]. Overall, the transformer module in our framework is crucial for identifying regulatory elements like enhancers, which depend on both local motifs and broader sequence context.

For the detailed mathematical formulations and technical aspects of the layer composition and normalization, please refer to Appendix B.

### 3.6. Proximal Policy Optimization for Feature Reduction

Proximal Policy Optimization (PPO) is an advanced reinforcement learning algorithm that effectively balances exploration and exploitation, ensuring stable and efficient training [25]. In our study, we use PPO to optimize the selection of informative features from the fused representations generated by the ResNet and Transformer modules, aiming to enhance downstream classification performance.

#### 3.6.1. Problem Formulation

The feature selection process can be framed as a sequential decision-making problem, where each decision involves keeping or discarding a specific feature. The PPO agent is trained to maximize the classification accuracy on the training dataset by selectively masking the features, and refining the input for the final classification layer.

#### 3.6.2. Environment Setup

The environment for deploying the PPO algorithm is structured as follows:**State**: The fused feature from residual and transformer modules, averaging it along the batch dimension, and then using it to represent the current observation.**Action**: A binary vector where each element indicates whether a specific feature is kept (1) or discarded (0).**Reward**: The reward is based on the classification accuracy achieved using the selected features, and is calculated as follows:
(5)$$R=\frac{N\prime }{N}$$Here, the reward *R* is defined as the ratio of correct predictions to the total number of samples in the batch. Specifically, $N^{\prime }$ denotes the number of correct predictions made after applying the feature mask generated by the PPO agent to the fused feature representation, while *N* represents the batch size. This reward signal helps guide the PPO agent in selecting the most informative features for the classification task.**Done**: The end of an episode, typically after processing all batches of the dataset.

#### 3.6.3. Training Procedure

The PPO algorithm iteratively updates the policy network by interacting with the environment, collecting experiences, and optimizing the policy based on the experiences. The training process involves the following steps:**Interaction with the Environment**: The agent observes the current state and selects an action (feature mask) based on its policy. The selected features are used to perform the classification task.**Reward Calculation**: The reward, defined as the classification accuracy on the training dataset, is computed and used to update the agent’s understanding of the effectiveness of the selected features.**Policy Update**: The agent’s policy is updated using the PPO objective, which aims to improve the expected reward while ensuring that the policy update does not deviate too much from the previous policy:
(6)$$L^{\mathrm{CLIP}}\left(\theta \right)=E_{t}\left[\min\left(r_{t}\left(\theta \right){\hat{A}}_{t},\mathrm{clip}\left(r_{t}\left(\theta \right),1-\epsilon ,1+\epsilon \right){\hat{A}}_{t}\right)\right]$$In this equation, $r_{t}\left(\theta \right)=\frac{\pi_{\theta }\left(a_{t}|s_{t}\right)}{\pi_{\theta_{old}}\left(a_{t}|s_{t}\right)}$ represents the ratio of the probability of taking action $a_{t}$ under the current policy $\pi_{\theta }$ to the probability under the old policy $\pi_{\theta_{old}}$. The term ${\hat{A}}_{t}$ is the advantage estimate, which measures how much better an action is compared to the expected value. The clipping function, $\mathrm{clip}(r_{t}\left(\theta \right),1-\epsilon ,1+\epsilon )$, ensures that the policy update remains within a specified range, controlled by the hyperparameter ϵ (typically set between 0.1 and 0.2), to maintain stability during training.**Episode Termination**: An episode ends when all batches in the dataset are processed. The environment is reset, and the next episode begins.

#### 3.6.4. Integration with Feature Extraction Modules

The PPO agent operates in conjunction with the ResNet and Transformer modules, which provide the fused feature vector. The agent’s actions dynamically adjust this vector by masking the less informative features. This integration ensures that the model focuses on the most critical features, improving accuracy and enhancing interpretability by highlighting the most relevant features for the classification task.

### 3.7. DeepEnhancerPPO

Here, we detail the design and workflow of our model, DeepEnhancerPPO, as illustrated in Figure 11.

#### 3.7.1. Model Design

The model comprises three main modules: the feature extraction module, the feature reduction module, and the classification module. These modules work synergistically to process DNA sequence data for two enhancer classification tasks: category and strength.

The feature extraction module integrates a simplified residual network with one-dimensional convolution (ResNet1D-18) and a transformer to capture a wide range of patterns within DNA sequences. The ResNet1D-18 module excels at extracting local and hierarchical features due to its convolutional operations and residual connections. These features range from simple motifs to complex fragments. In contrast, the transformer module specializes in capturing temporal relationships, positional dependencies, and long-range interactions within the sequence, providing a comprehensive embedding.

Both modules utilize embedded feature representations of DNA sequences as input, producing two-dimensional feature matrices through flattening and averaging operations, respectively. These feature matrices are then concatenated along the second dimension to create a fused feature map, which is subsequently averaged along the first dimension to generate the fusion feature vector, also known as the state vector, used as input for the PPO agent in the feature reduction module. The PPO agent applies a feature mask based on its decision policy to reduce the dimensionality of the fused features. This masking process emphasizes the most salient features necessary for accurate prediction while simultaneously reducing computational complexity.

The classification module receives the reduced feature vector and produces the final prediction. The prediction ACC serves as the reward for the PPO agent, which uses this feedback to optimize its feature reduction policy. During the training phase, the PPO agent is trained by optimizing the clipped surrogate objective Function (Equation 6), while the feature extraction and classification modules are trained by minimizing the binary cross-entropy loss function:(7)$$L\left(\hat{y},y\right)=-\frac{1}{N}\sum_{i=1}^{N}\left[y_{i}\log\left({\hat{y}}_{i}\right)+\left(1-y_{i}\right)\log\left(1-{\hat{y}}_{i}\right)\right]$$
where *N* is the total number of samples, $y_{i}$ is the true label of the *i*-th sample (0 or 1), and ${\hat{y}}_{i}$ is the predicted probability of the *i*-th sample being positive.

This comprehensive workflow enables DeepEnhancerPPO to effectively utilize both local and hierarchical features within DNA sequences, leveraging advanced deep learning and reinforcement learning techniques to achieve high accuracy in enhancer prediction.

#### 3.7.2. Training

Integrating reinforcement learning algorithms, such as PPO, into a complex model like DeepEnhancerPPO requires careful handling to ensure stable and effective training. To this end, we devised a three-phase training strategy:**Pre-training the Feature Extraction Modules:**The initial phase focuses on pre-training the feature extraction modules of DeepEnhancerPPO. This involves the ResNet1D-18 and transformer components, which are trained to provide accurate and meaningful features for subsequent processing. During this phase, the feature reduction module (PPO agent) is essentially disabled by initializing the feature mask sequence to all ones, ensuring that all features are utilized. The parameters of the ResNet1D-18 and the transformer encoder are optimized by minimizing the binary cross-entropy loss Function (Equation 7).**Pre-training the Feature Reduction Module:**In the second phase, we pre-train the feature reduction module, which is the PPO agent. Here, the parameters of the feature extraction (ResNet1D-18 and transformer encoder) and classification modules are frozen. The PPO agent interacts with the environment, collects experiences, and receives rewards based on classification accuracy, allowing it to learn an effective feature reduction policy. The PPO agent’s policy is updated by minimizing the clipped surrogate objective Function (Equation 6).**Joint Training:**In the final phase, we jointly train the feature extraction, feature reduction, and classification modules (see Algorithm 1). This involves simultaneously optimizing the binary cross-entropy loss for the classifier and the PPO objective function for feature reduction. Within each epoch, the feature extraction and classification modules are trained over multiple batches, followed by a single update of the PPO agent. This strategy ensures that DeepEnhancerPPO maintains stability and converges effectively during the training process.

**Algorithm 1** Joint Training Procedure for DeepEnhancerPPO
  1:**for** epoch=1 to epochs **do**  2:      **for** each batch in data_loader **do**  3:            Perform forward pass to extract features  4:            current_state ← model.get_feature_vector()  5:            action ← ppo.predict(current_state)  6:            model.apply_feature_mask(action)  7:            predictions ← model(batch)  8:            Optimize feature extraction and classification modules  9:      **end for**10:      Train PPO agent using experiences from the current epoch11:
**end for**



The application of PPO for feature reduction simplifies the input to the classifier and highlights the most salient features necessary for accurate predictions. This not only reduces computational load but also enhances model interpretability.

## 4. Conclusions

In this study, we introduce DeepEnhancerPPO, a novel end-to-end deep learning framework that uniquely integrates ResNet, transformer architectures, and reinforcement learning via PPO. This integration targets the dual objectives of robust enhancer classification and model interpretability. The ResNet and transformer modules are pivotal for extracting dense, distributed representations of k-mer words, where ResNet captures local and hierarchical features, while the transformer focuses on capturing long-range contextual dependencies.

A key innovation of our approach is the application of PPO for feature dimensionality reduction, enhancing computational efficiency and model interpretability by emphasizing the most relevant features for the classification tasks. Our experimental results confirm that DeepEnhancerPPO not only achieves superior performance across all evaluated metrics but also consistently outperforms existing models in enhancer classification, where each module within DeepEnhancerPPO proves to be indispensable for its overall performance.

Despite these successes, there are areas for further improvement. Future work will focus on enhancing performance in a broader range of DNA sequence classification tasks, particularly those involving variable sequence lengths, as in the benchmark proposed by Grešová et al. [52]. Additionally, improvements in embedding representation, such as leveraging pre-trained embeddings [29], or deeper feature extraction techniques, may further benefit the model.

## Figures and Tables

**Figure 1 ijms-25-12942-f001:**
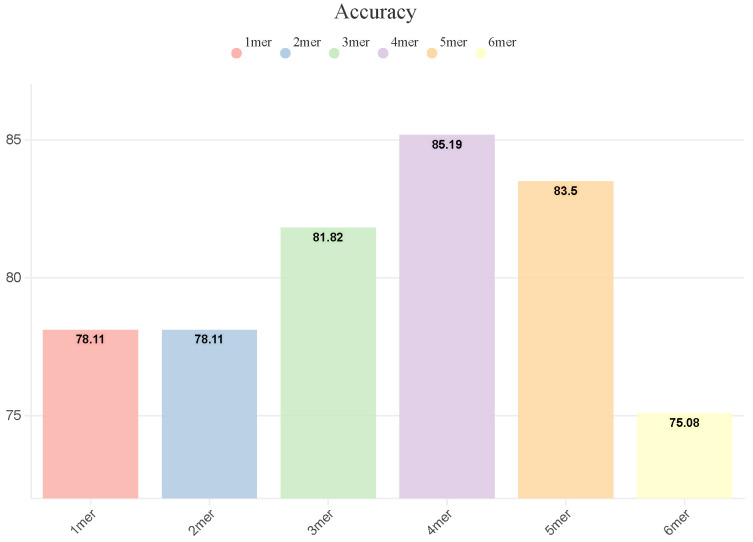
Accuracy of DeepEnhancerPPO with various k-mers on the validation dataset.

**Figure 2 ijms-25-12942-f002:**
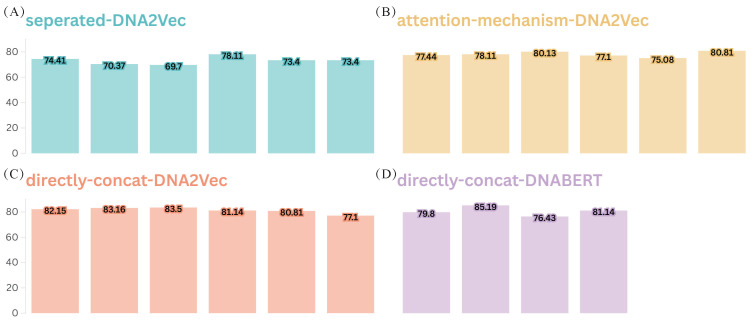
Accuracy of four different combination strategies for incorporating pre-trained embeddings, DNA2Vec and DNABERT, with the learnable 4-mer embedding. Panels (**A**–**C**) denote three different combination approaches for incorporating DNA2Vec embeddings into DeepEnhancer, where each bar represents a different value of k used for k-mer-based DNA2Vec embedding, ranging from 3 to 8. Panel (**D**) represents the performance of introducing DNABERT into DeepEnhancer, using the ‘direct-concat’ approach. The y-axis shows the accuracy of the validation dataset.

**Figure 3 ijms-25-12942-f003:**
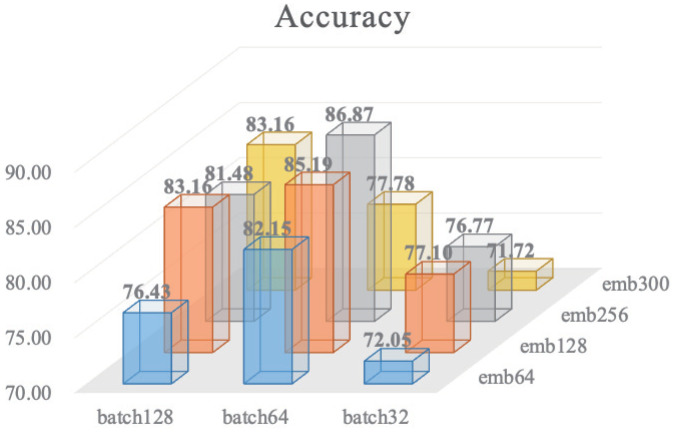
All potential combinations for batch size with embedding dimension. The embedding dimension set = [64, 128, 256, 300] and batch size set = [32, 64, 128]. The Z-axis value refers to the accuracy of the validation dataset.

**Figure 4 ijms-25-12942-f004:**
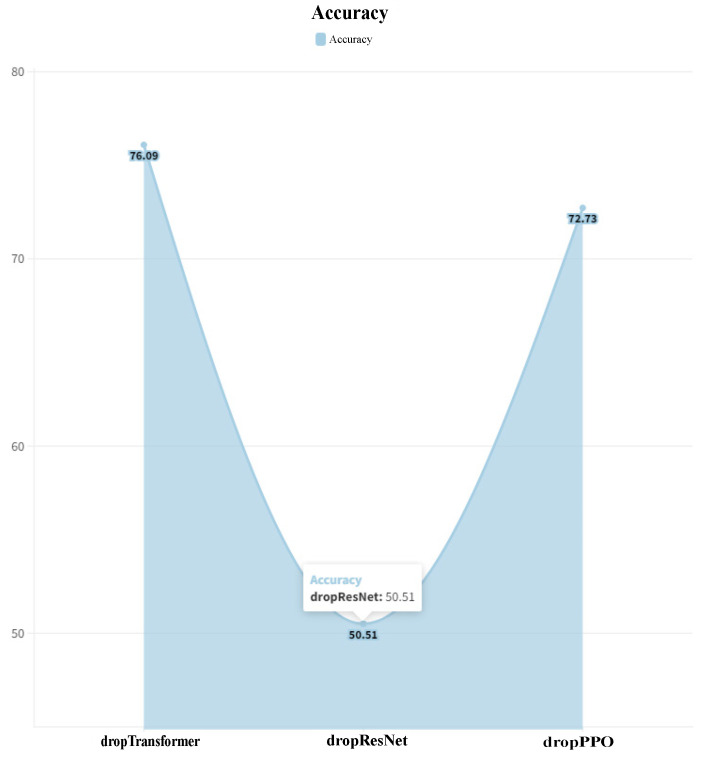
Ablation study results for DeepEnhancerPPO in the enhancer category classification task. dropTransformer denotes the Transformer module being removed; dropResNet and dropPPO refers to the removal of the ResNet and PPO modules, respectively. The Y-axis indicates the accuracy of the validation dataset after dropping the respective modules.

**Figure 5 ijms-25-12942-f005:**
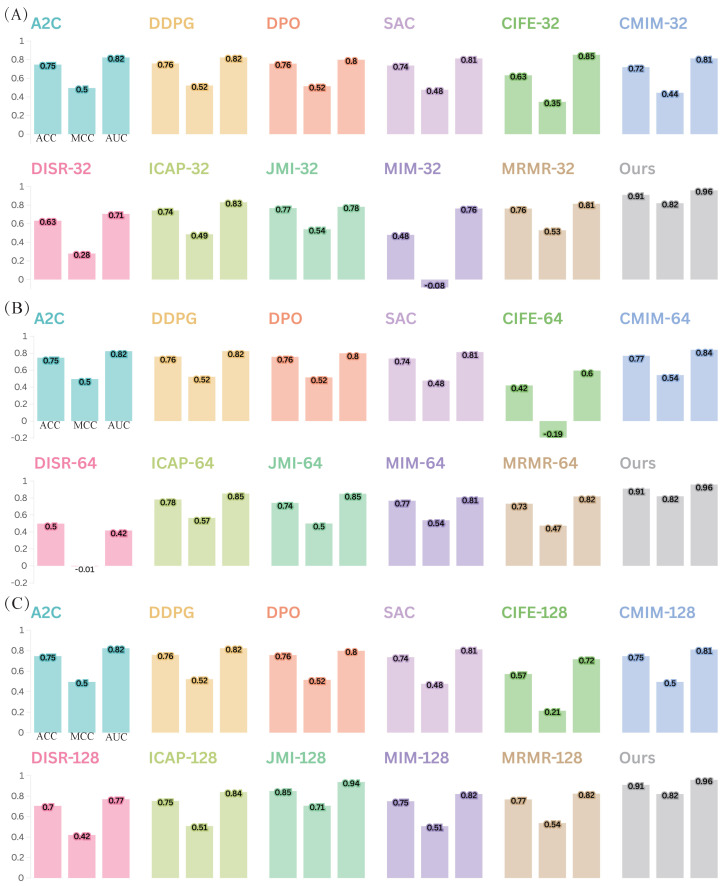
Comparison of DeepEnhancerPPO with 11 other feature reduction strategies on the independent test dataset for the first enhancer classification task: category classification, evaluated using three integrated metrics (ACC, MCC, and AUC). Panels (**A**–**C**) correspond to different numbers of top sub-features, specifically selecting the top 32, 64, and 128 sub-features, respectively.

**Figure 6 ijms-25-12942-f006:**
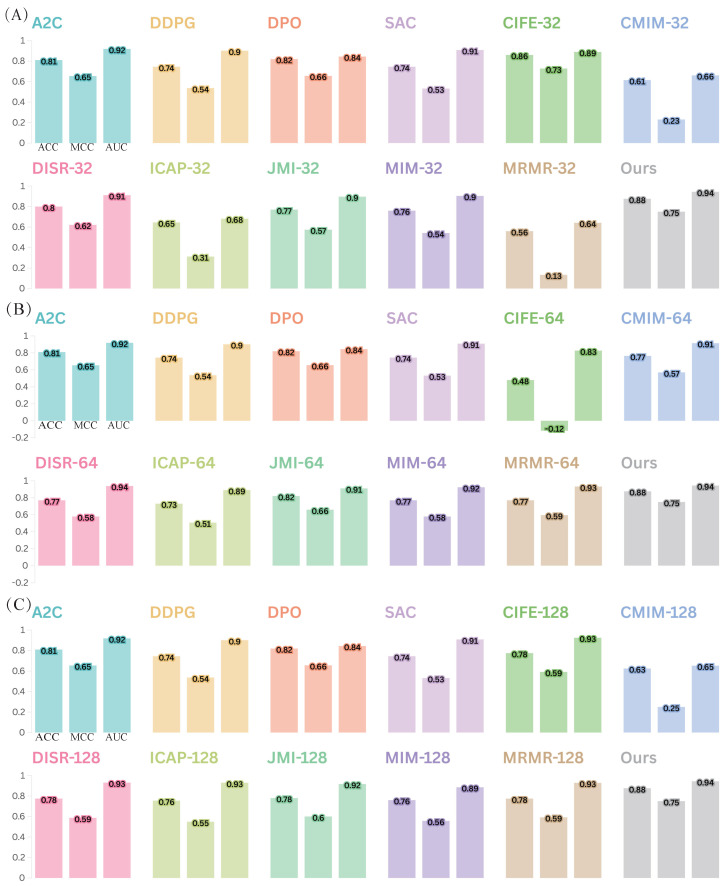
Comparison of DeepEnhancerPPO with 11 other feature reduction strategies on the independent test dataset for the second enhancer classification task: strength classification, assessed using three performance metrics (ACC, MCC, and AUC). Panels (**A**–**C**) illustrate the results for different numbers of top sub-features, specifically selecting the top 32, 64, and 128 sub-features, respectively.

**Figure 7 ijms-25-12942-f007:**
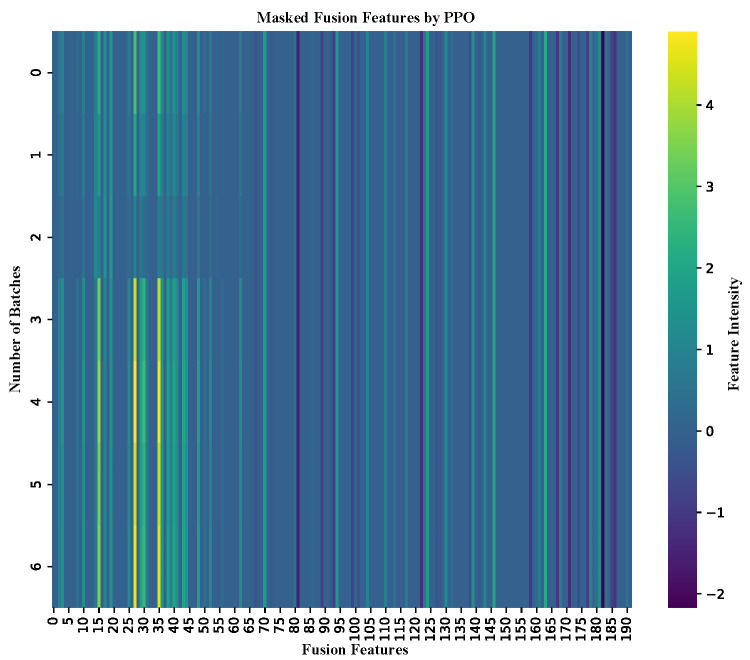
Heat map of masked fusion features for enhancer categorization on the independent dataset. The x-axis represents the 192 dimensions of the fusion feature set, with the intensity of colors indicating feature significance.

**Figure 8 ijms-25-12942-f008:**
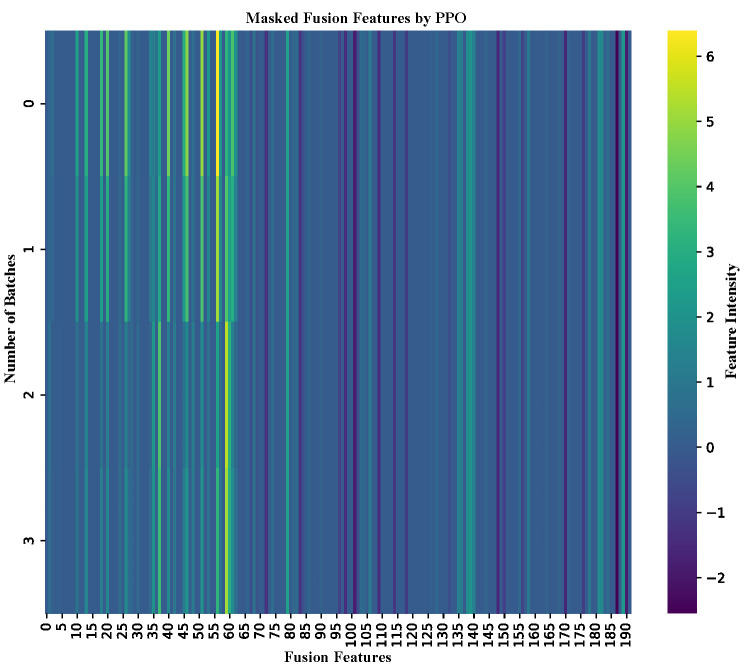
Heat map of masked fusion features for enhancer strength classification on the independent dataset. The x-axis represents the 192 dimensions of the fusion feature set, with color intensity indicating the relative importance of each feature.

**Figure 9 ijms-25-12942-f009:**
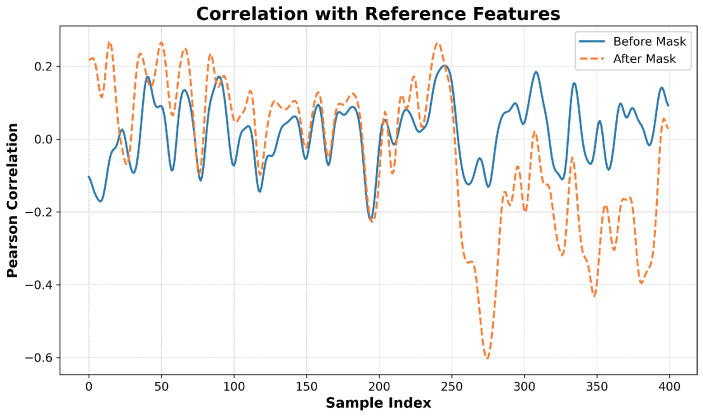
Correlation analysis of the before and after masked fusion features by PPO with the reference features, which consist of ‘GC Content’, ‘CpG Islands’, and ‘nucleotide composition’. The y-axis represents the Pearson correlation coefficient, and the x-axis denotes the sample index from the independent dataset.

**Figure 10 ijms-25-12942-f010:**
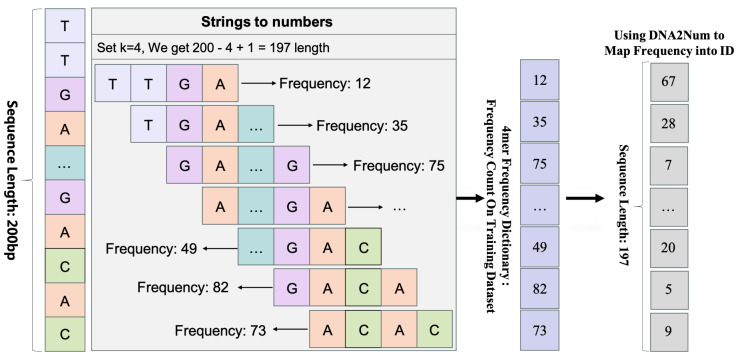
An illustration of k-mer splicing in a DNA sequence. The example shows how a 200 bp DNA sequence is split into 4-mers.

**Figure 11 ijms-25-12942-f011:**
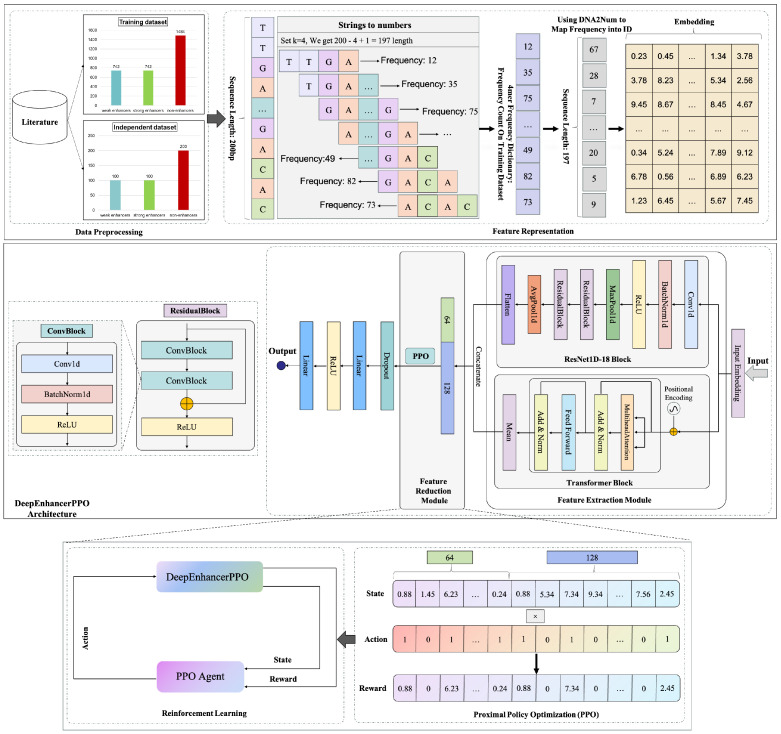
The complete workflow of DeepEnhancerPPO, illustrating the integration of ResNet, Transformer, and PPO for enhancer classification.

**Table 1 ijms-25-12942-t001:** Comparison of DeepEnhancerPPO with 24 state-of-the-art models on the independent datasets for five metrics on the first enhancer classification task: category classification.

Model	SN	SP	ACC	MCC	AUC
iEnhancer-2L [15]	0.7100	0.7500	0.7300	0.4604	0.8062
EnhancerPred [16]	0.7350	0.7450	0.7400	0.4800	0.8013
iEnhancer-EL [34]	0.7100	0.7850	0.7475	0.4964	0.8173
iEnhancer-5Step [35]	0.8200	0.7600	0.7900	0.5800	-
DeployEnhancer [36]	0.7550	0.7600	0.7550	0.5100	0.7704
iEnhancer-ECNN [18]	0.7520	0.7850	0.7690	0.5370	0.8320
EnhancerP-2L [37]	0.7810	0.8105	0.7950	0.5907	-
iEnhancer-CNN [38]	0.7825	0.7900	0.7750	0.5850	-
iEnhancer-XG [39]	0.7400	0.7750	0.7575	0.5150	-
Enhancer-DRRNN [40]	0.7330	0.8010	0.7670	0.5350	0.8370
Enhancer-BERT [22]	0.8000	0.7120	0.7560	0.5140	-
iEnhancer-RF [41]	0.7850	0.8100	0.7975	0.5952	0.8600
spEnhancer [42]	0.8300	0.7150	0.7725	0.5793	0.8235
iEnhancer-EBLSTM [43]	0.7550	0.7950	0.7720	0.5340	0.8350
iEnhancer-GAN [44]	0.8110	0.7580	0.7840	0.5670	-
piEnhPred [45]	0.8250	0.7840	0.8040	0.6099	-
iEnhancer-RD [44]	0.8100	0.7650	0.7880	0.5760	0.8440
iEnhancer-MFGBDT [46]	0.7679	0.7955	0.7750	0.5607	-
Enhancer-LSTMAtt [19]	0.7950	0.8150	0.8050	0.6101	0.8588
iEnhancer-Deep [47]	0.8150	0.6700	0.7402	0.4902	-
Rank-GAN [48]	0.7487	0.7563	0.7525	0.5051	0.8062
iEnhancer-DCLA [49]	0.7800	0.7850	0.7825	0.5650	0.8269
iEnhancer-BERT [50]	-	-	0.7930	0.5850	0.8440
ADH-Enhancer [51]	0.8420	0.8700	0.8430	0.6860	-
DeepEnhancerPPO (Ours)	0.9000	0.9200	0.9100	0.8202	0.9585

**Table 2 ijms-25-12942-t002:** Comparison of two versions of DeepEnhancerPPO with 24 state-of-the-art models on independent datasets for five metrics in the second enhancer classification task: strength classification.

Model	SN	SP	ACC	MCC	AUC
iEnhancer-2L [15]	0.7100	0.7500	0.7300	0.4604	0.8062
EnhancerPred [16]	0.7350	0.7450	0.7400	0.4800	0.8013
iEnhancer-EL [34]	0.5400	0.6800	0.6100	0.2222	0.6801
iEnhancer-5Step [35]	0.7400	0.5300	0.6350	0.2800	-
DeployEnhancer [36]	0.8315	0.4561	0.6849	0.3120	0.6714
iEnhancer-ECNN [18]	0.7910	0.7480	0.6780	0.3680	0.7480
EnhancerP-2L [37]	0.6829	0.7922	0.7250	0.4624	-
iEnhancer-CNN [38]	0.6525	0.7610	0.7500	0.3232	-
iEnhancer-XG [39]	0.7000	0.5700	0.6350	0.2720	-
Enhancer-DRRNN [40]	0.8580	0.8400	0.8490	0.6990	-
Enhancer-BERT [22]	-	-	-	-	-
iEnhancer-RF [41]	0.9300	0.7700	0.8500	0.7091	0.9700
spEnhancer [42]	0.9100	0.3300	0.6200	0.3703	0.6253
iEnhancer-EBLSTM [43]	0.8120	0.5360	0.6580	0.3240	0.6880
iEnhancer-GAN [44]	0.9610	0.5370	0.7490	0.5050	-
piEnhPred [45]	0.7000	0.7500	0.7250	0.4506	-
iEnhancer-RD [44]	0.8100	0.7650	0.7880	0.5760	0.8440
iEnhancer-MFGBDT [46]	0.7255	0.6681	0.6850	0.3862	-
Enhancer-LSTMAtt [19]	0.9900	0.8000	0.8950	0.8047	0.9637
iEnhancer-Deep [47]	0.7300	0.4900	0.6100	0.2266	-
Rank-GAN [48]	0.7068	0.6889	0.6970	0.3954	0.7702
iEnhancer-DCLA [49]	0.8700	0.6900	0.7800	0.5693	0.8226
iEnhancer-BERT [50]	-	-	0.7010	0.4010	0.8120
ADH-Enhancer [51]	0.8730	0.7500	0.8750	0.7740	-
DeepEnhancerPPO (Ours)	0.9500	0.6700	0.8100	0.6458	0.8524
DeepEnhancerPPO-Refined (Ours)	0.8900	0.8600	0.8750	0.7503	0.9442

**Table 3 ijms-25-12942-t003:** Comparison of DeepEnhancerPPO with two baselines on the same test dataset from a more recent benchmark for enhancer category classification. The comparison is made across five metrics for the first enhancer classification task: category classification.

Model	SN	SP	ACC	MCC	AUC
Baseline-1 [52]	-	-	0.6890	-	-
Baseline-2 [52]	-	-	0.8110	-	-
DeepEnhancerPPO (Ours)	0.9299	0.7500	0.8482	0.6738	0.8856

## Data Availability

The source code is openly accessible at https://github.com/Mxc666/DeepEnhancerPPO.git (accessed on 1 June 2024).

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
