# Peer review of "DeepEnhancerPPO: An Interpretable Deep Learning Approach for Enhancer Classification"

_ijms, 2024, doi:10.3390/ijms252312942_

Round 1

Reviewer 1 Report

Comments and Suggestions for Authors

The authors propose an innovative model named “DeepEnhancerPPO” for enhancer classification, integrating deep learning techniques, particularly ResNet, Transformer, and proximal policy optimization (PPO). The model aims to classify enhancer categories and regulatory strengths while maintaining high interpretability. Through evaluation metrics such as accuracy, Matthews correlation coefficient (MCC), and area under the curve (AUC), the model reportedly achieves impressive performance, surpassing 24 other models on enhancer category classification. The study highlights how the PPO component improves interpretability by selectively retaining essential features. However, there are some aspects where revisions are recommended:

1.     The selected hyperparameters, such as embedding dimension and batch size, need more extensive justification. While some reasoning is given, further elaboration on the impact of these choices on model generalization and computational efficiency would be valuable.

2.     The role of PPO in feature reduction is a compelling aspect of this study, yet it lacks depth in the interpretability analysis. Visualizing masked features is insightful, but more quantitative analysis or examples showing how PPO enhances interpretability for specific enhancer types would add robustness to the claims.

3.     For enhancer strength classification, DeepEnhancerPPO does not outperform all methods. The discussion would benefit from exploring reasons for these limitations and whether modifications to the current framework could improve performance in strength classification.

4.     While PPO is effective, other reinforcement learning techniques might also benefit feature reduction. A preliminary comparison with alternative methods could strengthen the case for PPO as the optimal choice.

5.     While the conclusion mentions potential future work, such as expanding the dataset, the authors could outline specific next steps, such as exploring additional regulatory elements or optimizing model components further.

Author Response

\textbf{Reviewer #1}

\textbf{Question 1:} The selected hyperparameters, such as embedding dimension and batch size, need more extensive justification. While some reasoning is given, further elaboration on the impact of these choices on model generalization and computational efficiency would be valuable.

\textbf{Response:} Thank you for your valuable feedback. In response to your suggestion, we have expanded our justification for the selection of hyperparameters, particularly embedding dimension and batch size. We conducted a hypothesis test, specifically McNemar's test, to evaluate the significance of the performance differences across hyperparameter combinations. Additionally, we recorded the computational time for each configuration to better highlight the trade-off between performance and efficiency. These details are now included in the revised manuscript on page 13, lines 489-505.

\textbf{Question 2:} The role of PPO in feature reduction is a compelling aspect of this study, yet it lacks depth in the interpretability analysis. Visualizing masked features is insightful, but more quantitative analysis or examples showing how PPO enhances interpretability for specific enhancer types would add robustness to the claims.

\textbf{Response:} Thank you for your insightful suggestion. In response, we have expanded the interpretability analysis of DeepEnhancerPPO. In addition to the discussion on the features masked by PPO to identify the most relevant ones, we have conducted a correlation analysis comparing the before and after masked features with reference features derived from well-established biological knowledge related to enhancer activity. These reference features include 'GC Content', 'CpG Islands', and 'nucleotide composition'. This analysis adds further depth to the interpretability of our model and strengthens our claims regarding the role of PPO in feature selection. The detailed changes are included in the revised manuscript on pages 18-21.

\textbf{Question 3:} For enhancer strength classification, DeepEnhancerPPO does not outperform all methods. The discussion would benefit from exploring reasons for these limitations and whether modifications to the current framework could improve performance in strength classification.

\textbf{Response:} Thank you for your valuable comment. In response, we have refined the hyperparameters for the second task (enhancer strength classification), in addition to using the same combination of hyperparameters as in the first task (enhancer category classification). This refinement led to an improvement in performance, positioning DeepEnhancerPPO as the second-best performing model for enhancer strength classification. This demonstrates the robustness and potential of DeepEnhancerPPO for both enhancer category and strength classification tasks. We have further elaborated on this refinement and its impact on performance in the revised manuscript, specifically on pages 14-16.

\textbf{Question 4:} While PPO is effective, other reinforcement learning techniques might also benefit feature reduction. A preliminary comparison with alternative methods could strengthen the case for PPO as the optimal choice.

\textbf{Response:} Thank you for your insightful suggestion. In response, we have conducted a comprehensive evaluation comparing PPO with seven classical feature selection algorithms and four state-of-the-art (SOTA) reinforcement learning (RL) techniques. This comparison demonstrates the superior performance of PPO for feature reduction. The detailed results of this evaluation can be found on pages 17-18, with the corresponding visualizations presented in Figures 7 and 8.

\textbf{Question 5:} While the conclusion mentions potential future work, such as expanding the dataset, the authors could outline specific next steps, such as exploring additional regulatory elements or optimizing model components further.

\textbf{Response:} Thank you for your valuable suggestion. In response, we have revised the Conclusion section to provide a more detailed description of future work. Specifically, we outline next steps focused on improving DeepEnhancer’s performance under varying conditions. You can find the updated discussion in the revised manuscript on page 22.

Reviewer 2 Report

Comments and Suggestions for Authors

The authors propose DeepEnhancerPPO, an deep learning model aimed at improving enhancer classification by incorporating ResNet and Transformer architectures for comprehensive feature extraction from DNA sequences. This model employs Proximal Policy Optimization (PPO), a reinforcement learning technique, to enhance classification accuracy and model interpretability by selectively retaining only the most relevant features. The methodology appears sound, and the model’s goals are both compelling and potentially impactful; however, the authors should address the following minor concerns to further validate their approach and strengthen the study’s contributions:

1)     Could the authors provide further clarification on how the PPO module enhances interpretability within the model framework? How does the model’s interpretability compare quantitatively with other models? Could the authors elaborate on specific interpretability metrics?

2)     Adding to this, are there any other visualizations obtained for interpretability other than the 2 heatmaps provided in the manuscript?

3)     When it comes to feature selection, it is unclear whether the authors have compared their method to other established feature selection (FS) schemes. If such comparisons have been made, presenting these results would be valuable in assessing the effectiveness of their approach relative to existing FS methodologies.

4)     In Figure 4, different accuracy scores are presented for various k-mer values. Some differences in accuracy scores between k-mer values appear quite close (for instance, between 77.44 and 78.11). Could the authors clarify whether these observed differences are statistically significant? Additionally, was any statistical test employed to substantiate the significance of these differences?

5)     Could the authors discuss the limitations observed in enhancer strength classification? Would adjusting hyperparameters between tasks improve performance?

Author Response

\textbf{Reviewer #2}

\textbf{Question 1:} Could the authors provide further clarification on how the PPO module enhances interpretability within the model framework? How does the model’s interpretability compare quantitatively with other models? Could the authors elaborate on specific interpretability metrics?

\textbf{Response:} Thank you for your insightful suggestion. In response, we have expanded the interpretability analysis of DeepEnhancerPPO. In addition to the discussion on the features masked by PPO to identify the most relevant ones, we have conducted a correlation analysis comparing the before and after masked features with reference features derived from well-established biological knowledge related to enhancer activity \cite{xiong2018genome}. These reference features include 'GC Content', 'CpG Islands', and 'nucleotide composition'. This analysis enhances the interpretability of our model by showing how PPO contributes to identifying biologically relevant features, further strengthening our claims regarding its role in feature selection. The detailed revisions can be found in the updated manuscript on pages 18-21.

\textbf{Question 2:} Adding to this, are there any other visualizations obtained for interpretability other than the 2 heatmaps provided in the manuscript?

\textbf{Response:} Thank you for your question. A similar answer to this can be found in our response to Question 1. In addition to the two heatmaps, we have conducted a correlation analysis between the before and after masked features and reference features related to enhancer activity, which provides further insights into the interpretability of the model. These additional analyses help reinforce our claims about the role of PPO in feature selection.

\textbf{Question 3:} When it comes to feature selection, it is unclear whether the authors have compared their method to other established feature selection (FS) schemes. If such comparisons have been made, presenting these results would be valuable in assessing the effectiveness of their approach relative to existing FS methodologies.

\textbf{Response:} Thank you for your insightful suggestion. In response, we have conducted a comprehensive evaluation comparing our PPO-based feature selection method with seven classical feature selection algorithms and four state-of-the-art (SOTA) reinforcement learning (RL) techniques. The results from this comparison clearly demonstrate the superior performance of PPO for feature reduction. The detailed results and corresponding visualizations of this evaluation can be found on pages 17-18, with the relevant figures presented in Figures 7 and 8.

\textbf{Question 4:} In Figure 4, different accuracy scores are presented for various k-mer values. Some differences in accuracy scores between k-mer values appear quite close (for instance, between 77.44 and 78.11). Could the authors clarify whether these observed differences are statistically significant? Additionally, was any statistical test employed to substantiate the significance of these differences?

\textbf{Response:} Thank you for your question. In fact, Figure 4, which explores the impact of pretrained embeddings, including DNA2Vec and DNABERT, is intended to assess whether these embeddings improve performance compared to the solely available learned 4-mer embedding, which achieves an accuracy of 85.19\% as shown in Figure 3. The comparisons within Figure 4 are not intended to be evaluated in isolation, as all results should be compared to the best performance shown in Figure 3 (using the 4-mer embedding with 85.19\%). We believe this comparison more accurately reflects the effectiveness of incorporating pretrained embeddings. The relevant details have been revised and can be found in section 3.1 on pages 10-12 of the manuscript.

\textbf{Question 5:} Could the authors discuss the limitations observed in enhancer strength classification? Would adjusting hyperparameters between tasks improve performance?

\textbf{Response:} Thank you for your insightful comment. In response, we have refined the hyperparameters for the second task (enhancer strength classification), in addition to using the same hyperparameters as in the first task (enhancer category classification). This refinement led to an improvement in performance, positioning DeepEnhancerPPO as the second-best performing model for enhancer strength classification. This highlights the flexibility and potential of DeepEnhancerPPO for both enhancer category and strength classification tasks. We have provided additional details on this refinement and its impact on performance in the revised manuscript, specifically on pages 14-16.

Reviewer 3 Report

Comments and Suggestions for Authors

In this study, Mu et al. propose a novel deep learning architecture named DeepEnhancerPPO for enhancer classification.  The model combines ResNet and Transformer modules to extract diverse features from DNA sequences.  It treats DNA sequences as natural language sentences and uses k-mer techniques for sequence splitting and embedding. The model shows superior performance compared to 24 state-of-the-art models in enhancer category classification. However, before consider publishing in our journal, several aspects are required to improve the quality of manuscript:

Major Comments:

1.    The DNA2Vec is not the state-of-the-art model, please use more advanced pre-trained models, such as DNAbert and DNABERT2, in embedding methods for comparison.

2.    The authors mentioned that the reward is calculated based on the classification accuracy, but did not elaborate on how the reward is specifically calculated and the complete form of the reward function.

3.    Please use DPO to achieve feature dimensionality reduction.

4.    There is a lack of performance evaluation on other benchmarks.

5.    In the K-mer encoding section, the authors show the results under different k-mer, however, they lack the analysis and discussion of this result.

6.  It seems that the authors used the work of ML visual for the model structure diagram. It is recommended to cite it or state this in the acknowledgments.)

7.  As described in the manuscript, there are so many (up to 24) previously-published models for enhancer classification. Please discuss why and how your method can improve the accuracy in the prediction? Is this only because of using proximal policy optimization?

Author Response

\textbf{Reviewer #3}

\textbf{Question 1:} The DNA2Vec is not the state-of-the-art model, please use more advanced pre-trained models, such as DNAbert and DNABERT2, in embedding methods for comparison.

\textbf{Response:} Thank you for your valuable suggestion. In response, we have expanded our evaluation to include the state-of-the-art pre-trained model, DNABERT \cite{ji2021dnabert}, in combination with our learned 4-mer embedding, in addition to DNA2Vec. This comparison allows us to assess the impact of more advanced embeddings on the performance of DeepEnhancerPPO. The details of this evaluation can be found in Section 3.1, on pages 10-12 of the revised manuscript.

\textbf{Question 2:} The authors mentioned that the reward is calculated based on the classification accuracy, but did not elaborate on how the reward is specifically calculated and the complete form of the reward function. 

\textbf{Response:} Thank you for your comment. We have provided a more detailed explanation of the reward function used by PPO for feature reduction in the revised manuscript. Specifically, we have clarified how the reward is calculated based on classification accuracy, and the complete form of the reward function is now included on pages 6-7 of the manuscript.

\textbf{Question 3:} Please use DPO to achieve feature dimensionality reduction.

\textbf{Response:} Thank you for your valuable suggestion. In response, we have conducted a comprehensive evaluation comparing our PPO-based feature selection method with seven classical feature selection algorithms and four state-of-the-art (SOTA) reinforcement learning (RL) techniques, including the recent Direct Policy Optimization (DPO) algorithm proposed by \cite{rafailov2024direct}. The results from this comparison demonstrate the superior performance of PPO for feature reduction. The detailed results and corresponding visualizations of this evaluation can be found on pages 17-18, with relevant figures presented in Figures 7 and 8.

\textbf{Question 4:} There is a lack of performance evaluation on other benchmarks.

\textbf{Response:} Thank you for your insightful suggestion. In response, we have further evaluated the performance of DeepEnhancerPPO on a more recent benchmark dataset \cite{grevsova2023genomic}, specifically focusing on enhancer category classification, to demonstrate the generalizability and robustness of our proposed approach. The details of this evaluation are provided in the revised manuscript on pages 3 and 16-17.

\textbf{Question 5:} In the K-mer encoding section, the authors show the results under different k-mer, however, they lack the analysis and discussion of this result.

\textbf{Response:} Thank you for your insightful suggestion. In response, we have expanded the discussion regarding the impact of different k-mer values. Specifically, we now provide a deeper analysis of the results obtained with various k values for k-mer encoding. You can find the detailed discussion in the revised manuscript on page 10.

\textbf{Question 6:} It seems that the authors used the work of ML visual for the model structure diagram. It is recommended to cite it or state this in the acknowledgments.)

\textbf{Response:} Thank you for your insightful suggestion. We did reference the style of ML Visual for the model structure diagram; however, we created the diagram ourselves based on this reference. In line with your advice, we have now acknowledged their work in the acknowledgments section. You can find the relevant acknowledgment in the revised manuscript on page 22.

\textbf{Question 7:} As described in the manuscript, there are so many (up to 24) previously-published models for enhancer classification. Please discuss why and how your method can improve the accuracy in the prediction? Is this only because of using proximal policy optimization?

\textbf{Response:} Thank you for your insightful question. We have addressed this point in the ablation study section, where we discuss the contributions of the ResNet, Transformer, and PPO modules. As shown in Figure 6, the most significant impact on DeepEnhancerPPO’s performance comes from the ResNet module, as evidenced by a 34.68\% drop in validation accuracy upon its removal (from 85.19\% to 50.51\%). Figure 6 further highlights that while all three modules—ResNet, Transformer, and PPO—are essential for DeepEnhancerPPO’s success, their relative importance differs. Specifically, ResNet is the most significant contributor, followed by PPO, with the Transformer module also positively impacting enhancer category classification. For further details, please refer to the revised manuscript on pages 13-14.

Round 2

Reviewer 3 Report

Comments and Suggestions for Authors

The authors have well replied my comments, I recommend accept in current form.